# AutoSP: Unlocking Long-Context LLM Training Via Compiler-Based Sequence Parallelism

**Ahan Gupta***, **Zhihao Wang***, **Neel Dani**
SSAIL Lab, University of Illinois Urbana-Champaign
Champaign, IL, USA
`{ag82, zhihaow6, neeld2}@illinois.edu`

**Masahiro Tanaka**
Anyscale
San Francisco, CA, USA
`mtanaka@anyscale.com`

**Olatunji Ruwase**
Snowflake
San Mateo, CA, USA
`tunji.ruwase@snowflake.com`

**Minjia Zhang**
SSAIL Lab, University of Illinois Urbana-Champaign
Champaign, IL, USA
`minjiaz@illinois.edu`

## Abstract

Large-language-models (LLMs) demonstrate enormous utility in long-context tasks which require processing prompts that consist of tens to hundreds of thousands of tokens. However, existing LLM training libraries do not provide easy to use abstractions to optimize for long-context training, instead focusing on optimizations for models with large parameter counts through ZeRO-3/FSDP, Tensor and Pipeline parallelism. This forces users to rewrite LLM training libraries to incorporate compositions of various complex long-context optimizations, such as sequence-parallelism, to training pipelines; a process that requires in-depth expertise, reducing developer productivity. To tackle these challenges, we introduce AutoSP: the first automated solution to automatically optimize LLM training for longer-contexts. AutoSP compiles models and applies a targeted set of optimizations: automated sequence parallelism, and long-context aware activation-checkpointing, to drastically enhance LLM trainability at negligible cost to throughput. Our evaluation demonstrates AutoSP's capability on both NVIDIA and AMD hardware, increasing training contexts by upto $2.7\times$ and $2.5\times$ respectively over competitive hand-written baseline at negligible cost to runtime performance.

## 1 Introduction

Large Language Models (LLMs) are increasingly being trained with long-context data for scenarios such as document understanding (Han et al., 2025; Zhu et al., 2025; Landeghem et al., 2023), multi-step reasoning (Cobbe et al., 2021; Feng et al., 2020), and extended multi-turn dialogue generation (Touvron et al., 2023; Bai et al., 2024). These use cases often contain input sequences ranging from tens to hundreds of thousands of tokens, creating massive activation memory demands and pushing the memory and system limits of GPU clusters.

To circumvent out-of-memory errors, researchers have explored Sequence Parallelism (SP), a key enabler for long-context training. State-of-the-art SP strategies such as DeepSpeed-Ulysses (Jacobs et al., 2024) and RingAttention (Liu et al., 2023) distribute the sequence dimension of activations across devices and allow the training engine to leverage aggregated GPU memory to train longer contexts with increasing device counts.

Despite effectively enabling long-context training, existing SP are implemented in eager mode and tightly coupled to specialized training frameworks such as DeepSpeed (Rajbhandari et al., 2020) and Megatron-LM (Shoeybi et al., 2020). Integrating SP to new training pipelines typically requires invasive code refactoring, which makes it difficult to apply across diverse model architectures and hardware platforms. Developers must manually insert communication collectives (e.g., `all2all`) between operators that require the full input sequence (such as attention), manage activation layouts

---

*Equal contribution.

across devices, and ensure correctness in both forward and backward passes. These manual efforts reduce scientists' productivity and limit portability.

To improve productivity, researchers have begun to lift several complex distributed training strategies such as ZeRO-3/FSDP (Rajbhandari et al., 2020) into SoTA deep-learning compilers, e.g. PyTorch-2.0 (Ansel et al., 2024). Examples include: SimpleFSDP (Zhang et al., 2024) & DeepCompile (Tanaka et al., 2025), which implement ZeRO-3/FSDP as a series of compiler passes in the PyTorch-2.0 ecosystem. However, each of these techniques focuses on how to increase model parameter counts, uncovering different ways to shard model parameters rather than explicitly optimize for long-context training. While these efforts successfully lift data and model parallelism into compiler abstractions, they do not address parallelism strategies tailored to long-context training. This raises the question: can SP also be lifted into a deep-learning compilation stack to enable automated sequence parallelism?

In this work, we focus on lifting SP into PyTorch-2.0's compiler ecosystem as a compiler-based, PyTorch-native implementation. However, achieving this introduces several challenges. First, PyTorch-2.0's compilation pipeline includes multiple intermediate-representations (IRs) such as Torch-IR, Aten-IR, and Inductor-IR, to name a few. Each IR operates at a different abstraction level and encodes the input program at varying levels of granularity. This makes identifying an appropriate IR to conduct program-analysis, so as to recover the necessary information to apply semantically-preserving rewrites that transform single-GPU code into distributed sequence-parallel execution, a major challenge. A fine-grained abstraction

```
1 # Register passes #
2 reg_passes(['auto_sp, sp_ac'])
3 # Initialize dist training #
4 dist.init(SP_GROUP_SIZE)
5 # Compile model. #
6 model.compile()
7 # model training #
8 for batch in data_set:
9     output = model(batch[:,SP_GROUP])
10    output.backward()
11    optimizer.step(model)
```

Listing 1: AutoSP provides an easy to use interface to apply long-context optimizations to training pipelines. It manages activation layouts between devices, inserts communication collectives and ensures correctness of the forward and backward training pass.

tion will make program-analysis (to uncover important attributes about the input model) increasingly non-trivial whilst a coarse-grained abstraction will make semantic-rewrites (to insert communication collectives, transform buffer sizes and recompute manually indexed tensors) infeasible. Second, inferring sequence-dependent tensor shapes for resizing intermediate buffers is non-trivial to do within a compiler. (1) The lowering process inserts intermediate data-movement operators (such as transpositions), frequently changing the sequential axis of buffers. (2) SP strategies require resizing the sequential axis of only certain buffers (such as token and position id buffers), leaving other buffers (such as attention masks) untouched. Consequently, disambiguating which token's sequential axis requires resizing is challenging. Third, lifting SP into the PyTorch-2.0 compilation stack has consequences to other optimization passes that PyTorch-2.0 natively supports, notably: activation-checkpointing (AC). AC also enables memory savings for training by discarding activations in the forward pass and rematerializing them in the backwards pass to compute gradients. However, naively rematerializing activations together with SP triggers extraneous communication in the backwards pass, adversarially impacting runtime performance.

To tackle these challenges, we introduce AutoSP, the first compiler-based, PyTorch-native implementation of sequence parallelism. AutoSP introduces two key components: (1) a sequence-parallel transformation pass that automatically inserts communication collectives and reshapes activations, and (2) a sequence-aware activation checkpointing pass that exploits the compute-memory characteristics of long-context training. With just a few lines of code (as shown in 1), scientists can compile standard PyTorch models into distributed long-context training pipelines that scale input lengths without manual engineering. Our evaluation demonstrates that AutoSP significantly improves trainability at negligible loss to training speed on diverse hardware backends (NVIDIA and AMD GPUs), enabling training with up to $2.7\times$ longer input context lengths compared to hand-written SP implementations such as DeepSpeed-Ulysses and RingAttention.

## 2 BACKGROUND

**Sequence parallel training.** Sequence parallelism (SP) is a key enabler for long-context training. These strategies enable scaling input sequence lengths with increasing GPU resources by

sharding input tensors and activations across the sequence dimension. Communication collectives are inserted in the forward and backward pass as necessary to correctly shuffle tokens to the desired device. A popular SP strategy, and the focus of this work, is DeepSpeed-Ulysses (Ulysses) (Jacobs et al., 2024). We illustrate how Ulysses operates with 3-SP groups in Fig. 1.

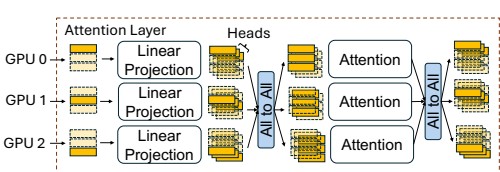

First, tokens are sharded across the sequence dimension, with different devices operating on different parts of the input sequence. Next, linear projections form multiple Q/K/V heads. Since linear projections operate pointwise across the sequential dimension, each device directly operates on its own tokens with no additional communication. However, since attention requires the entire input-context, an all-to-all reshuffles tokens, re-sharding the activations across the head-dimension. Each device locally computes attention on its respective head(s) after which another all-to-all reshuffles the tokens back to their original input sizes, re-sharding across the sequence dimension.

Figure 1: DeepSpeed-Ulysses with 3-SP groups. `alltoall` operators toggle the layout of activations at attention-layer boundaries. Linear-projections operate on the partial sequence length, while attention-layers operate on a subset of the heads.

**PyTorch-2.0 compiler.** PyTorch-2.0 (Ansel et al., 2024) is a just-in-time deep-learning compiler that targets training workloads. It comprises of two components: dynamo and inductor each with a series of *compiler passes* that progressively lowers and optimizes code. An example code-snippet of a neural-network and its progressive lowering through different IRs, is shown in Fig. 2.

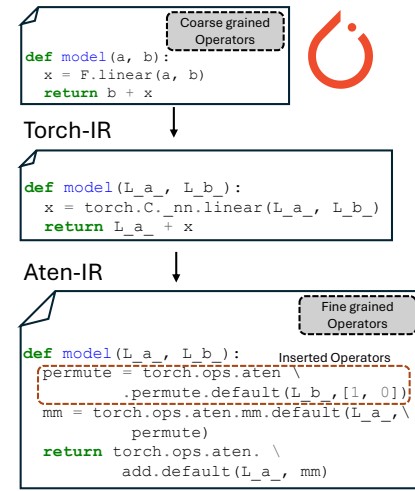

*Dynamo.* The input to dynamo is a model comprising of PyTorch & python operators. Dynamo then executes the function and records a *trace*, represented as its intermediate representation: a computation graph. Each node in the computation-graph comprises of Torch-IR statements, which loosely correspond to statements in the original input program. Next, AOTAutograd lowers each Torch-IR statement to Aten-IR statements, which consist of finer-grained operators. Aten-IR statements do not consist of higher-level abstractions such as linear or attention layers, but instead consist of (batch) matrix-multiplication, convolution, and data-movement operators. Each Torch-IR statement is accordingly lowered to its corresponding set of (multiple) Aten-IR statement(s), forming an *FX-graph*. For example, in Fig. 2, we observe that the `linear` operator in Torch-IR is lowered to two operators in Aten-IR: `permute` and `mm`. At this stage, a variety of compiler-

Figure 2: A sample neural network compiled using PyTorch-2.0. We illustrate the lowering to Torch-IR and Aten-IR that occur within Dynamo and the extra operators inserted during the lowering process.

passes to optimize the FX-graph are applied, notably automated acitvation-checkpointing (AC). The AC compiler-pass (Chillee) is responsible for selecting which tensors to rematerialize in the backwards pass without incurring performance penalties. It reduces this problem to a network-flow construction whose min-cut determines the tensors to rematerialize. We describe it in detail in Section 3.2.

*Inductor.* Finally, inductor consumes the output Aten-IR FX-graph and lowers it to a custom define-by-run IR, subsequently code-generating necessary kernels specialized to the backend microarchitecture.

## 3 AUTOSP

AutoSP lifts SP parallelism as a compiler-pass into the PyTorch-2.0 compiler stack to optimize long-context training. Fig. 3 is an overview of how AutoSP's compiler-passes interoperate with the PyTorch-2.0 compilation stack to optimize LLM training code. Section 3.1 describes how we enable

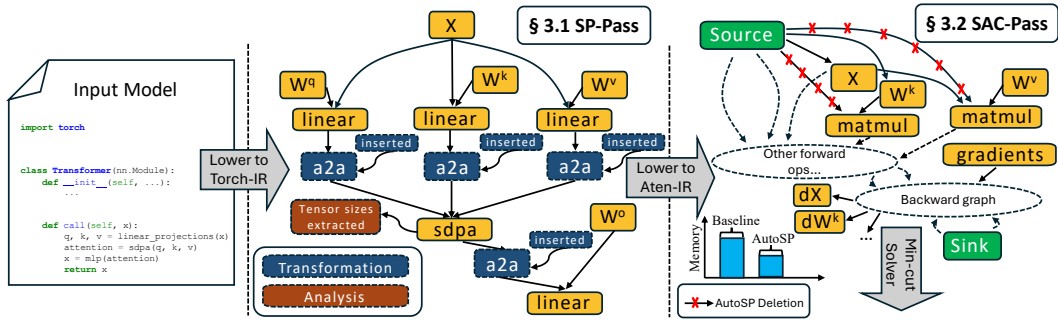

Figure 3: An overview of AutoSP. AutoSP enables an automated approach to scale input context lengths for long-context training through a targeted set of compiler optimization passes: an automated sequence-parallelism pass (Section 3.1), and a SP-aware long-context AC-pass: SAC (Section 3.2).

automated sequence parallelism as a compiler pass, and Section 3.2 describes our long-context aware activation checkpointing strategy.

## 3.1 AUTOMATED SEQUENCE PARALLELISM

**Challenges.** To implement automated sequence parallelism, we must select (a single) PyTorch IR(s) to analyze and transform accordingly. However, recovering the necessary model information through program analysis and subsequently transforming the model through semantically-preserving rewrites is non-trivial for three reasons. (1) Each inserted communication collective must have token-buffers instantiated to a particular size parameterized by the model-dimension, batch-size and sequence length. However, these parameters are not explicitly represented in any IR. (2) Existing intermediate buffers need to be resized to different shapes depending on the SP group-size as well as their placement within the neural network. For example, in DeepSpeed-Ulysses, buffers within the attention layer should be resized to operate on the full-sequence but a subset of all the heads, whilst buffers within MLP layers operate on the full model-dimension but on a partial sequence-length. (3) Certain operations require manual indexing for correctness, e.g. indexing the causal mask to appropriately apply to attention matrices, and need to be automatically recomputed.

To tackle these challenges, AutoSP's SP-pass accordingly analyzes computational structures to extract the pertinent information required for transforming single-GPU code to distributed sequence-parallel code. We next describe how it operates.

### 3.1.1 ANALYSIS AND TRANSFORMATIONS

Our pass operates in two stages. First, we analyze tensor sizes to gather information about the batch, sequence and hidden dimension that parameterize the model. Next, we transform the IR by: (1) inserting communication collectives at appropriate places in the network with appropriately sized buffers to store their output, (2) adjusting the sizes of existing buffers within the graph to account for sequence sharding. (3) Recompute any manually indexed tensors appropriately. All our analysis and transformations operate on Torch-IR.

**Why AutoSP analyzes and transforms Torch-IR?** Our SP-pass operates on Torch-IR as the analysis and transformations are significantly more challenging to accomplish on Aten-IR for three reasons. (1) Torch-IR more closely resembles the neural-network programmed by the user (see Section 2) with operators such as linear and attention-layers, making it easier to identify which parts of the graph belong to which layer for appropriate tensor resizing. On the other hand, equivalent operators in Aten-IR are represented as a series of finer-grained operators, such as mat-muls and permutations, making it challenging to reason about which operators belong to linear-projections and attention-operators respectively. (2) The lowering process from Torch-IR to Aten-IR inserts various data-layout transformations such as reshapes and permutes, obscuring information as to which dimension corresponds to the sequence, batch, and hidden sizes of a tensor, making it challenging to appropriately resize the correct tensor dimension. (3) Torch-IR operates on only the forward-pass resulting in our transformations operating on a simpler computation-graph, merely requiring each new added node

to have a registered conjugate gradient operator. On the other hand, Aten-IR operates on both the forward and backwards pass and requires more complex transformations to the computation-graph.

**Program analysis to uncover training parameters.** To correctly instantiate token-buffers for communication collectives, the correct batch, sequence and model-dimensions need to be extracted. Fortunately, we can analyze the input nodes of the entire computation graph to extract the necessary information. Since the input to the computation-graph is the data-loaded after preprocessing, it is *guaranteed* to resemble a particular shape depending on the problem domain. For example, in natural-language tasks, the data will be a `[batch, seq_length]`-sized tensor. Next, to acquire the model-dimension we traverse the graph until we encounter an attention operator and inspect its output ND tensor whose last two dimensions are: `[num_heads, head_dim]`-sized. The product of the outer two dimensions is the model-dimension.

```
1  # b=batch, s=seq, h=heads, d=dim
2  def transform(mod,b,s,h,d):
3   for node in mod.nodes:
4    # WS = World Size
5    part_seq = s/WS
6    # Resize buffers.
7    if node.name in RESIZE_BUFS:
8     if node.name in ATTN_OPS:
9      resize_attn(node, [b,s,h/WS,d])
10    else:
11     resize_others(node,
12              [b,part_seq,d])
13    # Recalculate manual indexing
14    if node.name in INDEX_OPS:
15     recalc_index(node, node.args)
16    # Insert comm. collectives
17    if node.name is ATTN_OPS[0]:
18     buf_proj = [batch,seq,heads/WS,
19              dim]
20     insert_before(node, all_to_all,
21              buf_proj)
22    elif node.name is ATTN_OPS[-1]:
23     buf_attn = [batch,seq/WS,heads,
24              dim]
25     insert_after(node, all_to_all,
26              buf_attn)
```

Listing 2: AutoSP's Transformation Pass: converting single-GPU code to sequence-parallel multi-GPU code. The pass resizes buffers, recomputes manually indexed tensors, and inserts communication collectives.

**Program transformation.** After acquiring the batch, sequence and model-dimensions, we have the necessary information to transform the computation graph from single-GPU to distributed sequence-parallel code. Listing 2 illustrates how we transform the existing computation-graph, `mod`, comprising of Torch-IR statements. We traverse through the graph, and for each node: (1) Check if it belongs to the `RESIZE_BUFS` set, and accordingly resize its buffers depending on its placement in the attention or linear-projection/MLP-layers. (2) Check if it belongs in the `INDEX_OPS` set, and accordingly resize its tensor indexing. (3) Check if it is the first/last attention-op and accordingly instantiate communication buffers and insert the necessary `alltoall` before/after the attention-layer. We manually curate the `RESIZE_BUFS`, `INDEX_OPS`, and `ATTN_OPS` sets by analyzing dynamo FX-graphs of compiled hand-written transformer implementations.

## 3.2 SEQUENCE-PARALLEL AWARE ACTIVATION CHECKPOINTING

**Challenges.** In addition to the SP-pass, activation checkpointing (AC) is an important memory reducing optimization that enables longer context training. `torch.compile` provides an automated AC-pass (Chillee) that operates on Aten-IR to compose with arbitrary neural networks. However, naively composing its AC-pass with AutoSP's SP-pass within the compiler-stack leads to sub-optimal performance for long-context training. We briefly explain how PyTorch-2.0's automated AC-pass functions, and why it is insufficient.

**PyTorch-2.0's AC-pass.** PyTorch's AC-pass operates on Aten-IR, within Dynamo. Its primary function is to reduce memory consumption of model training without incurring performance penalties. The optimization reduces the problem to a network-flow construction whose min-cut determines the tensors to rematerialize. We give an example code-snippet and its equivalent network-flow construction in Fig. 4. The input to the optimization is an FX-graph comprising of Aten-IR statements. First, a *joint-graph* of the forward and backwards graph is constructed. Next, the source node is connected to all the input tensors of the graph, and all the nodes reachable from the incoming gradients are connected to the sink. Then, capacities, representing costs, are assigned to each node. A node's capacity is determined by a heuristic function comprising of various characteristics such as the output activation memory produced. Finally, the problem is converted from a flow on nodes to a flow on edges. Each edge's capacity is set to `inf` and each node is split into two: an `_in` and `_out` node. Incoming edges to each node are connected to its respective `_in` node and outgoing edges are connected to its respective `_out` node. An edge with the original node's capacity connects `_in` to `_out`. A min-cut on this graph will cut only finite-capacity edges from a node's `_in` to `_out`

(highlighted by green circles in Fig. 4); only nodes on this cut are stored. Intuitively, the min-cut identifies the smallest cost set of activations to preserve to compute the necessary dependencies in the backwards pass. PyTorch-2.0 additionally enforces constraints on which nodes can be rematerialized, resulting in its conservative nature which we explain next.

**Why PyTorch-2.0's AC solution is insufficient.** PyTorch-2.0's AC-pass is effective in identifying which activations to store without compromising runtime performance in polynomial time. However, it makes the conservative decision to disallow rematerialization of many *seemingly* compute-intensive operators such as: mat-muls, and scaled mat-muls to name a few, based purely on operator type, without considering runtime cost in long-context settings. This is enforced by connecting each compute-intensive node's _in to the source with infinite capacity (indicated by the additional dotted red line in Fig. 4), enforcing that node, or some downstream value of it, to belong to the min-cut. However, in long-context training, we observe that certain compute-intensive operators take a small fraction of overall compute and can accordingly be rematerialized without incurring performance penalties. AutoSP's AC strategy exploits this. We explain these observations next.

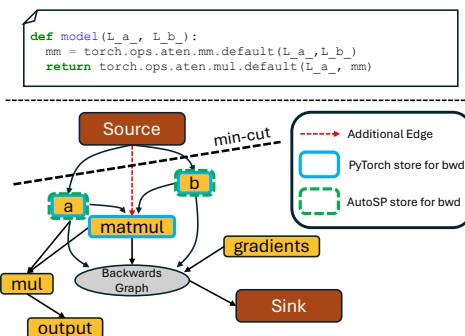

Figure 4: A comparison of PyTorch-2.0 vs. AutoSP's AC-passes on a sample code-snippet. The red line is the additional edge from source to node in PyTorch-2.0's AC-pass to enforce no rematerialization of compute heavy operators. AutoSP's AC-pass, in removing this constraint, reduces memory-consumption at negligible cost to throughput.

**Observations.** We analyze the structure of compute operations in modern LLMs to identify regions that can be appropriately rematerialized. For a transformer with: batch-size $b$, sequence-length $s$, number of heads $h$, head dimension $d$, and MLP hidden dimension $d_{ffn}$, we have that:

$$2bhs^2d \text{ Attention FLOPS}$$
$$8bhsd^2 \text{ Linear-projection FLOPS}$$
$$4bhsd_{ffn}d \text{ MLP FLOPS}$$

When training on long-contexts, we have that $s >> d, h, d_{ffn}$, which results in the following fraction of FLOPs linear-projection and MLP layers take over all the compute operations:

$$\frac{8bhsd^2 + 4bhsd_{ffn}d}{2bhs^2d + 8bhsd^2 + 4bhsd_{ffn}d} \approx O\left(\frac{1}{s}\right) \quad \text{as } s \to \infty \tag{1}$$

Indicating that the fraction of compute operations for linear-projection and MLP matrix-multiplications decreases as a function of input-sequence length. This observation underpins AutoSP's automated AC strategy.

**AutoSP's AC strategy.** AutoSP exploits equation 1, building upon PyTorch-2's automated AC strategy. However, instead of conservatively banning *every* compute-intensive operator, we permit configurations where (batch) matrix-multiplications and other compute-intensive operators outside of the attention layer are rematerialized. We achieve this by iterating over the joint-graph and removing any additional edges from the source to compute-heavy operators, resulting in only inputs to the graph (tensors a and b in Fig. 4) connecting to the source. We then dispatch this mutated joint-graph to PyTorch-2.0's AC strategy. This change enables traning on significantly longer context lengths at negligible cost to training throughput.

## 4 EVALUATION

We evaluate AutoSP with a comprehensive set of experiments. We demonstrate its effectiveness in enhancing trainibility of various models and sizes in Section 4.1, and detailed breakdowns of the impact of each component in Section 4.2.

**Setup.** We evaluate AutoSP and all the baselines on NVIDIA GH200-96GB & A100-80GB and AMD MI250-64GB hardware. All experiments use PyTorch-2.7 with CUDA 12.8 (on NVIDIA

GPUs), and ROCm 6.4 (on AMD GPUs). To implement AutoSP, we lift the DeepSpeed-Ulysses SP scheme into PyTorch-2.0's compilation stack and integrate all our compiler optimizations into the DeepSpeed project, due to its popularity in training large scale LLMs.

**Baselines.** We compare AutoSP to both compiler-optimized distributed training solutions and hand-optimized SP solutions to demonstrate the memory and compute efficiency of our approach. Specifically, we compare against ZeRO-3 (FSDP) Rajbhandari et al. (2020) optimized through `torch.compile()` in PyTorch-2.0, and hand-written DeepSpeed-Ulysses (Jacobs et al., 2024) & RingAttention implementations when compiled under PyTorch-2.0's inductor backend. We use DeepSpeed-Ulysses' code in the original DeepSpeed repository[1] and RingFlashAttention[2], which are both highly optimized implementations of SP strategies. We evaluate all techniques on a range of model sizes: Llama-3.2 1B & 3B, Llama-3.1 8B, and Llama-2 13B, covering models with either Grouped-Query-Attention (GQA) or Full-Attention.

## 4.1 MAIN RESULTS

In this section, we evaluate how AutoSP impacts model trainability: the maximum trainable sequence length prior to encountering OOM issues.

**Trainability.** For different techniques, we measure the maximum sequence length trainable prior to OOM on 8 NVIDIA A100-80GB in Fig. 5. For all models, we use ZeRO-1, setting the SP group-size to 2 and the DP group-size to 4 with the exception of the ZeRO-3 augmented with `torch.compile()` baseline. Compared to ZeRO-3 (FSDP), AutoSP enables training on upto 5×, 5.6× and 2.5× longer input sequences for the 3B, 8B and 13B models, respectively. The trainability gains come from AutoSP's compiler-based SP-pass, an optimization that targets long-context training unlike the ZeRO-3 baseline, which instead targets models with large parameter counts. Moreover, AutoSP achieves significant trainability gains compared to both inductor compiled DS-Ulysses and RingAttention implementations. Compared to DS-Ulysses, AutoSP enables training on longer input sequences by upto 2.14×, 3× and 1.88× for the 3B, 8B and 13B models, respectively. Compared to RingAttention, AutoSP enables training on upto 2.14×, 3× and 1.6× for the 3B, 8B and 13B models, respectively. The additional gains over hand-written SP implementations come from the SP-aware AC-pass that exploits equation 1, rematerializing compute-heavy operators (such as mat-muls) for low runtime costs but large memory gains. The trainability gains are especially pronounced for the 8B model compared to 3B as many of the compute-heavy operators, such as linear-projections and MLPs, produce activations parameterized by the model hidden-dimension. These need to be stored to compute gradients in the backwards pass and result in larger models producing significantly more activation memory due to compute-heavy operators. AutoSP, in rematerializing these large tensors, alleviates memory issues at negligible runtime cost. The trainability gains are less pronounced for the 13B model as optimizer states begin to consume a substantial portion of memory (~50%).

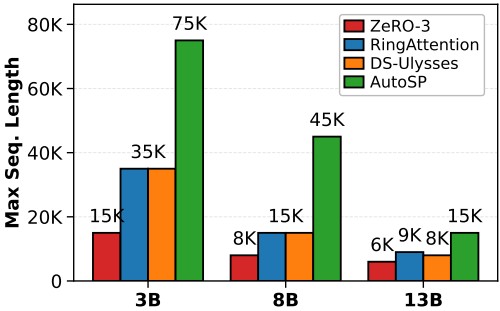

Figure 5: Maximum sequence length prior to OOM across various model sizes. AutoSP increases the trainability of all model sizes.

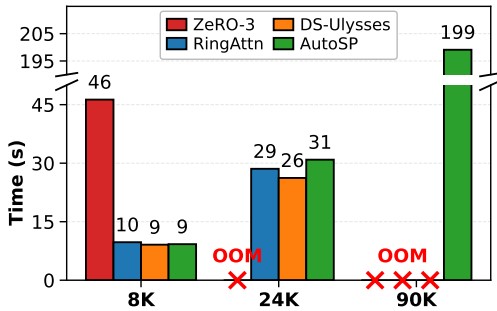

Figure 6: End-to-End training iteration times (over 10 iterations) for Llama-3.1 8B on 8 GPUs. AutoSP maintains similar per-training iteration time whilst significantly increasing trainability.

---

[1]https://github.com/deepspeedai/DeepSpeed/tree/master/deepspeed/sequence

[2]https://github.com/zhuzilin/ring-flash-attention

**Runtime Performance.** For different techniques, we identify the impact on training iteration times at various sequence lengths in Fig. 6. We focus on a Llama-3.1 8B model, using ZeRO-1 with a fixed DP-size of 2 and SP-size of 4, and evaluate on 8 GPUs, measuring time taken for 10 training iterations. We toggle the sequence-aware AC pass on for AutoSP only if OOM issues occur. Compared to the ZeRO-3 baseline compiled with `torch.compile()`, AutoSP reduces per-iteration times by 5× whilst increasing trainability by an order of magnitude. ZeRO-3 introduces more communication collectives over ZeRO-1 with SP, resulting in slower per-iteration times. Moreover, compared to RingAttention and DS-Ulysses, AutoSP increases trainability by 3.75× with negligible cost to runtime-performance. Finally, we observe that DS-Ulysses is faster than RingAttention due to RingAttention's p-step (where p is the SP size) communication latency which exchanges key/values in a

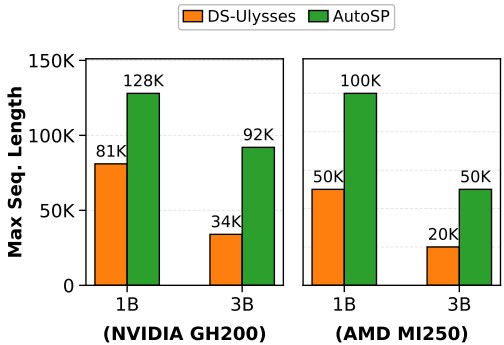

Figure 7: Comparing the max sequence length prior to OOM across different hardware. AutoSP enables training on longer sequences on NVIDIA superchips and AMD hardware.

ring-like pattern across SP-groups. Comparitively, DS-Ulysses introduces a single all-to-all to re-shard token-buffers. In modern clusters, intra-node all-to-alls are fast due to pairwise communication links between devices.

## 4.2 ANALYSIS

In this section, we run additional studies to demonstrate the effectiveness of AutoSP, as well as to ascertain the impact of each optimization on different components of LLM training. We compare AutoSP with DS-Ulysses only (rather than RingAttention) given it enables training on similarly sized context-lengths whilst being slightly faster.

**Trainability on diverse hardware.** We run an additional trainibility study on different NVIDIA and AMD hardware, identifying the maximum sequence length before encountering an OOM in Fig. 7 to demonstrate AutoSP's portability. We focus on smaller, 1B and 3B models, running on either 2 GH200-96GB (NVIDIA) GPUs, or 2 AMD MI250-64GB GPUs. On NVIDIA hardware, AutoSP enables training on 1.58× (2.70×) longer sequence lengths on the 1B (3B) models respectively. On AMD hardware, AutoSP enables training on 2× (2.5×) longer input sequences on the 1B (3B) models, respectively. AutoSP consistently delivers significant trainability gains across diverse hardware, and model sizes through its targeted long-context optimizations.

**Runtime performance on diverse hardware.** For different techniques, we measure the per-iteration end-to-end training time (averaged across 100 training iterations) on NVIDIA (GH200-96GB) and AMD (MI250-64GB) hardware in Fig. 8, demonstrating the

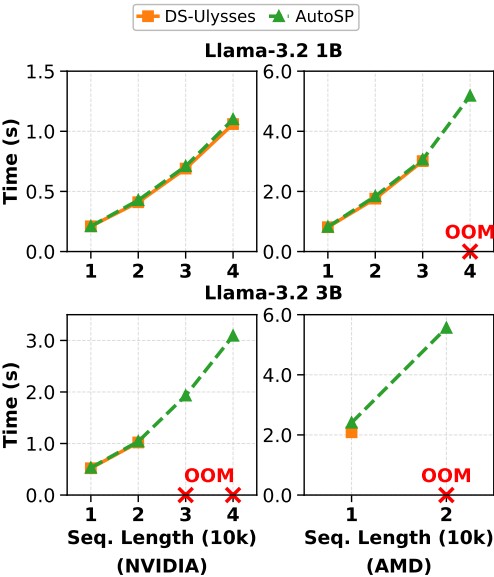

Figure 8: Average execution time of various Llama 3.2 model sizes at different sequence lengths on NVIDIA and AMD hardware. AutoSP matches the performance of hand-written baselines and supports longer sequence training.

performance-portability of our approach. We trigger the SP-aware AC-pass only to avoid OOM issues for AutoSP. On sequence lengths that all techniques can train on, AutoSP has the following speedups: 0.97× (1B) and 0.98× (3B) compared to the inductor baseline on NVIDIA hardware, respectively. On AMD hardware, AutoSP has the following speedups: 0.97× (1B) and 0.87× (3B) compared to the inductor baseline, respectively. We note two observations. Despite being a general and performance-portable compiler pass, AutoSP achieves 97% of DS-Ulysses' hand-written baseline

whilst providing an upto 2.7× trainability gain. Without AutoSP's targeted optimizations, training at long-contexts quickly becomes infeasible.

**Breakdown analysis.** We breakdown the impact of AutoSP's optimizations in Fig. 9 on a NVIDIA GH200-96GB. We breakdown the activation memory produced by the attention and MLP operators as well as the per-iteration runtime of the forward and backward passes when training a Llama-3.2 1B model using a sequence length of 40k. Overall, AutoSP reduces memory consumption of the attention and MLP operators by 13.03× and 2.22×, respectively. The marked impact on MLP operators arises due to the presence of many mat-muls, which AutoSP's AC-pass rematerializes to alleviate memory consumption at low runtime costs. On the other hand, AutoSP incurs a 1.14× cost runtime performance cost for the backward pass, whilst having similar forward pass times as a result of the extra rematerialized operators, which are recomputed in the backwards pass only.

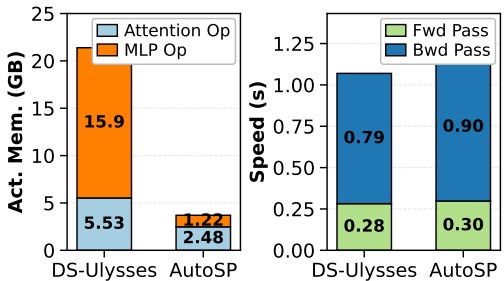

Figure 9: Breakdown of Attention and MLP operator memory consumption and forward iteration times. AutoSP reduces the activation memory of Attention and MLP operators with a marginal performance difference.

**Ablation study.** Finally, we toggle different optimizations on/off to demonstrate their impact on trainability and speed in Table 1. The top row indicates the maximum trainable input context length before an OOM, and the bottom row indicates the average training iteration time at a fixed 40k input sequence length. Overall, AutoSP's optimizations result in a 1.58× trainability gain whilst achieving 89% of DeepSpeed-Ulysses' per-iteration training time. Moreover, the incremental trainability gain of the AC-pass over the SP-pass is 1.66× with a mere 7% decrease in runtime-performance. This significant trainability gain with little runtime-performance cost is due to equation 1. At longer contexts, the attention-operator's FLOPs dominate runtime, enabling traditionally compute-heavy operators to be rematerialized with marginal performance costs and large memory gains. Note that the baseline is a highly hand-optimized SP implementation with several optimizations such as communication-computation overlapping using streams. Nevertheless, the SP-pass achieves 97% of the baseline's performance as a general compiler-pass.

Table 1: Various optimizations in AutoSP against a baseline on a Llama-3.1 1B. AutoSP supports training on significantly longer sequence lengths at minimal cost to performance.

| Method | | Max Tokens | Speed (s) |
|---|---|---|---|
| DS-Ulysses | | 81,000 | **1.06** |
| AutoSP | SP-Pass | 77,000 | 1.09 |
| | SP & AC-Pass | **128,000** | 1.19 |

## 5  RELATED WORK

**Parallel training strategies.** ZeRO-3/FSDP (Rajbhandari et al., 2020), Tensor Shoeybi et al. (2020) & Pipeline (Huang et al., 2019) parallelism are training strategies that target models with large parameter counts, reducing per-device memory consumption of optimizer, model, activation and gradient states. Expert-parallelism (Lepikhin et al., 2020; DeepSeek-AI et al., 2025; Dai et al., 2024) targets large sparse mixtures-of-experts which contain many intermediate expert MLPs. Though effective in enhancing the trainability of large-language model training, these parallel strategies do not explicitly target long-context training and are insufficient to scale input context lengths.

**Automated optimizations.** Deep-compile (Tanaka et al., 2025) provides an automated approach to implement ZeRO-3/FSDP using profile-guided optimization. Though effective, FSDP does not explicitly target long-context training. General-Single-Program-Multiple-Data (GSPMD) (Xu et al., 2021), is an automated parallelization strategy in XLA guided through user annotations, requiring some human effort. Lastly, deep-learning compilers such as: TVM (Chen et al., 2018), Mirage (Wu

et al., 2025), and AITemplate (Meta, 2022), focus on schedule rewrites for inference workloads only and do not consider inter-GPU parallelism, a key optimization for long-context training.

**Activation checkpointing.** Various works propose AC techniques (Jain et al., 2020; Kirisame et al., 2021; Chen et al., 2016). They primarily consist of two approaches. (1) Search-based optimization (e.g. via integer-linear-programming), which may not scale up to today's LLM sizes (billion parameter models). (2) Static-policies (e.g. checkpoint chunks of $\sqrt{N}$ layers' activations), which may result in extraneous communication calls in the backward pass in the SP-setting. Comparatively, our SP-aware AC-pass exploits observations of compute & memory properties of LLM training at long-contexts to alleviate memory consumption at negligible cost to runtime performance. Additionally, AutoSP's sequence-aware AC pass is implemented in PyTorch-2.0's compiler stack, requiring no human intervention. In contrast, existing AC strategies often require invasive code-rewrites.

## 6 CONCLUSION

In this paper, we present AutoSP, the first compiler-based, PyTorch-native solution for training large-language-models at long-contexts. Through a combination of automated sequence-parallelism (SP) and a sequence-aware AC strategy, AutoSP achieves significant sequence length extensions at negligible cost to training throughput. Our results demonstrate that compiler-driven, PyTorch-native automation provides a practical and portable foundation for long-context model training.

## REPRODUCIBILITY STATEMENT

We have made several efforts to ensure the reproducibility of our work. We have documented the critical components of our implementation in Section 3 and our evaluation in Section 4 additionally documents our setup with a description of our hardware and software versions. We have ensured that each result is consistent with good benchmarking practices, including taking the average over multiple runs. Moreover, all our code and benchmarks will be made publicly available to further enhance reproducibility.

## ACKNOWLEDGMENTS

We sincerely appreciate the anonymous reviewers. Their insightful feedback helps significantly improve the quality of the paper. This research was supported by the National Science Foundation (NSF) under Grant No. 2441601. The work utilized the Delta and DeltaAI system at the National Center for Supercomputing Applications (NCSA) and Jetstream2 at Indiana University through allocation CIS240055 from the Advanced Cyberinfrastructure Coordination Ecosystem: Services & Support (ACCESS) program, which is supported by National Science Foundation grants #2138259, #2138286, #2138307, #2137603, and #2138296. The Delta advanced computing resource is a collaborative effort between the University of Illinois Urbana-Champaign and NCSA, supported by the NSF (award OAC 2005572) and the State of Illinois. UIUC SSAIL Lab is supported by research funding and gift from IBM, Google, Amazon, and AMD.

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
