# OpenReview forum: "AutoSP: Unlocking Long-Context LLM Training Via Compiler-Based Sequence Parallelism"
_ICLR.cc/2026/Conference — ICLR 2026 Poster_

### Official Review · Reviewer_4FNs · 2025-10-28

**Soundness:** 4
**Presentation:** 3
**Contribution:** 3
**Rating:** 8
**Confidence:** 3

**Summary:**

The compilation of a PyTorch 2.0 model goes through multiple stages. Dynamo, the stage responsible for running the model for the first time and recording the computation graph, can be split into two passes: Torch-IR (a higher-level graph that still resembles model layers) and Aten-IR (a lower-level graph of primitive tensor ops, e.g. matrix multiplies, permutes, convolutions, data moves). The authors present a PyTorch compiler patch integrated into the DeepSeek library that implements two optimizations:

1. Takes the implementation of sequence parallelism from DeepSpeed-Ulysses into the Torch-IR pass, so that the attention layers can be automatically distributed across GPUs for parallel processing of long context sequences. The compilerized version works with arbitrary PyTorch models with minimal code changes.

2. Adds a layer of optimization to the Aten-IR pass, optimizing activation checkpointing (AC). They show that PyTorch’s stock AC is too conservative for long-sequence training because it forbids rematerializing classically compute-heavy ops (e.g., matmuls, convs). They show that as sequence length grows, MLP/linear matmuls constitute a vanishing fraction of total FLOPs, so these “heavy” ops can be rematerialized cheaply. By removing this restriction, they enable a longer context to fit into the memory.

They conduct an extensive empirical study, demonstrating that their implementation
- is compatible with different types of hardware (NVIDEA, AMD),
- allows long sequences to fit into memory when compared to ZeRO-3 and hand-written DS-Ulysses,
- Scales well with SP group size

**Strengths:**

The greatest value of this work is that it automates the implementation of SP for practitioners, requiring minimal code changes. This has the potential to accelerate research and development of language model architectures with a wide impact. The authors present an original contribution of an optimized activation checkpointing strategy and provide empirical results that demonstrate the effectiveness of the proposed changes. The paper is clearly written and provides sufficient documentation on the empirical experiments.

**Weaknesses:**

- The paper lacks low-level technical details about how AutoSP manipulates Torch-IR in complex models beyond toy examples. The authors should make sure this is addressed during the release of the source code.
- The paper overemphasizes the benefits of AutoSP without detailing potential downsides, such as the scenarios where the overhead of recomputation in AC becomes substantial.

**Questions:**

- Comparing AutoSP to other context-length extension strategies (e.g., HuggingFace Accelerate with Context Parallelism) would make the results more trustworthy

---

> ### Author Response · Authors · 2025-11-16
>
> We thank you for your thoughtful reviews, and are encouraged by your positive comments on the impact of our work and corresponding rigorous empirical assessment.
>
> **The paper lacks low-level technical details about how AutoSP manipulates Torch-IR in complex models beyond toy examples. The authors should make sure this is addressed during the release of the source code.**
>
> We thank you for this feedback, we will ensure that this is addressed in the release of the source code, and will evaluate a larger variety of models.
>
> **The paper overemphasizes the benefits of AutoSP without detailing potential downsides, such as the scenarios where the overhead of recomputation in AC becomes substantial.**
>
> We thank you for this suggestion. Upon ablating the impact of our SP-aware AC-pass on runtime performance, we observe that it incurs a 9% overhead for a corresponding 1.66x increase in training context-lengths. In practice, we have found this overhead to be tolerable as without our pass, training on longer contexts quickly becomes infeasible. Moreover, the AC-pass, in being a separate compiler pass, need only be triggered if OOM issues arise, resulting in this runtime overhead for sequence lengths only when existing SP solutions cannot feasibly train a LLM to completion.
>
> **Comparing AutoSP to other context-length extension strategies (e.g., HuggingFace Accelerate with Context Parallelism) would make the results more trustworthy**
>
> We thank you for the valuable suggestion. HuggingFace accelerate uses DeepSpeed as a backend, with RingAttention as its CP implementation. During the rebuttal process, we have additionally benchmarked against RingAttention (which should be equivalent to HuggingFace Accelerate's CP as we also use DeepSpeed). We give our results in the following table (the configurations are identical to those in figures 5 and 6).
>
> Comparing RingAttention against AutoSP across various model-sizes on 8 GPUs (with an identical configuration to Figure 5 in our paper):
>
> | Model Size | AutoSP | RingAttention |
> |------------|--------|---------------|
> | 3B         | 75k    | 35k           |
> | 8B         | 45k    | 15k           |
> | 13B        | 15k    | 9k            |
>
> On average, AutoSP incurs a 2.26x average increase in training contexts across all model-sizes in these settings.
>
> Comparing RingAttention against AutoSP across different GPU counts (with an identical configuration to Figure 6 in our paper):
>
> | GPU Count | AutoSP | RingAttention |
> |-----------|--------|---------------|
> | 4         | 30k    | 12k           |
> | 8         | 90k    | 37k           |
>
> On average, AutoSP incurs a 2.45x average increase in training contexts across all model-sizes in these settings.
>
> However, we note that RingAttention can be used when the SP group size exceeds the number of attention-heads, a scenario where Ulysses does not apply.

---

> > ### Author Response · Authors · 2025-11-23
> >
> > Dear Reviewer 4FNs,
> >
> > Thank you once again for the time and effort you’ve invested in reviewing our manuscript. We would like to kindly remind you that we have diligently addressed each point raised in your review. We would be more than happy to address any additional concerns or comments you may have.
> >
> > Thank you!
> >
> > Best Regards,
> >
> > Authors of AutoSP (Submission 22044)

---

> > > ### Author Response · Authors · 2025-11-27
> > >
> > > Dear Reviewer 4FNs,
> > >
> > > We hope you're doing well. We submitted our follow up responses on 11/15/2025, and we just wanted to gently check in since it has been 12 days and there are only a few days left in the rebuttal period.
> > >
> > > If there are any remaining concerns we can help clarify, we'd be very happy to provide additional explanations. Thank you again for the time and effort you've put into reviewing our paper.
> > >
> > > Best Regards,
> > >
> > > Authors of 22044

---

### Official Review · Reviewer_AhGd · 2025-10-30

**Soundness:** 3
**Presentation:** 3
**Contribution:** 1
**Rating:** 2
**Confidence:** 2

**Summary:**

This paper introduces AutoSP, a novel compiler-based system for PyTorch-2.0 that automates optimizations for training Large Language Models (LLMs) with very long contexts.

**Strengths:**

The article's writing is commendable, particularly for its very clear and accessible explanations of technical details.

**Weaknesses:**

1.  I am skeptical of the core claim that context parallelism is difficult to implement. An API-based approach, inspired by Flash Attention (e.g., `flash_attn_func(query, key, value)`), seems more suitable than integrating it to `torch.compile()`. While also a one-line modification, this API provides users with greater transparency and explicit control, rather than obscuring the underlying logic.

2.  The comparison between the proposed compile-stage activation checkpointing and traditional layer-wise checkpointing is vague. The paper primarily demonstrates advantages over a standard `torch.compile()` baseline but fails to clearly articulate its benefits over conventional layer-wise gradient checkpointing. This is a significant omission, as the latter is a common, effective, and simple-to-use practice in LLM training.

3.  Most critically, the ideas presented are neither groundbreaking nor particularly novel. They appear to be targeted optimizations for implementing a specific function more efficiently within an existing general purpose framework, rather than a substantive innovation. As such, this contribution seems more appropriate for a pull request to the PyTorch repository than a publication at ICLR.

**Questions:**

1.  Why was DeepSpeed Ulysses selected as the baseline? Does it exclusively support eager-mode, $O(N^2)$ attention? If so, it is an inappropriate baseline. Standard $O(N^2)$ attention is outdated; modern implementations using Flash Attention or PyTorch SDPA can easily train 16K contexts on a single 80GB GPU, especially with DeepSpeed ZeRO-3 enabled. A valid comparison would require using faster context-parallel (CP) implementations, such as Megatron-LM's CP or Ring Flash Attention [1].

2.  Using `torch.compile()` for activation checkpointing is uncommon in practice. Standard LLM implementations (e.g., in Hugging Face) wrap each layer with `torch.utils.checkpoint`. This standard, LLM-specific approach should be discussed. The official PyTorch API is not complex:

    ```python
    torch.utils.checkpoint.checkpoint(layer_function, *inputs, use_reentrant=False)
    ```
    Please clarify the efficiency difference between your activation checkpointing method and this standard `torch.utils.checkpoint` implementation.

[1] https://github.com/zhuzilin/ring-flash-attention

---

> ### Author Response · Authors · 2025-11-16
>
> We thank you for your thoughtful reviews, and are encouraged by your comments about the clarity and accessibility of our paper.
>
> **Skeptical of the core claim that context parallelism is difficult to implement.**
>
> We have addressed this in the common questions section of the rebuttal.
>
> **An API-based approach seems more suitable than integrating it to torch.compile().**
>
> First, an API-based approach still requires users to define manual SP groups and figure out how to manually compose such a technique with other training strategies, such as ZeRO/FSDP. Second, our approach figures out a better strategy for how SP should interact with AC, something an API-based approach for SP would not accomplish as APIs are written in isolation of other strategies. In doing so, we can drastically enhance the trainability (and compositionality with other strategies) of our method compared to existing SP-based approaches.
>
> **The comparison between the proposed compile-stage activation checkpointing and traditional layer-wise checkpointing is vague. Please clarify the efficiency difference between our activation checkpointing and standard checkpointing via torch.utils.checkpoint.**
>
> Traditional layer-wise approaches such as [1], [2] require manual instrumentation of the code using torch.utils.checkpoint APIs. Moreover, these strategies are not inherently sequence aware and do not support longer-context training with minimal impact to training throughput. Instead, such strategies are designed to train larger models (akin to the comparison between ZeRO and SP). Our checkpointing strategy, on the other hand, requires no manual effort and is specially curated towards training longer contexts.
>
> Additionally, our AC approach, in operating on finer-grained Aten-IR operators, exposes a larger search space of more performant schedules that trade-off trainability for performance. Traditional layer-wise AC is applied to a specific layer, that means if we apply checkpointing to the MLP-layer in a transformer *all* the activations within that layer are discarded. Though effective in reducing memory usage, it is difficult to extract reasonable performance given that it operates on coarse-grained layers such as MLP & Attention-layers. By circumventing this and operating directly on Aten-IR, which consist of individual mat-muls, sigmoids, etc… operators themselves, the space of expressible AC-schedules is larger compared to using torch.utils.checkpoint.
>
> [1] https://arxiv.org/pdf/1604.06174
>
> [2] https://arxiv.org/pdf/2205.05198
>
> **The ideas presented are neither groundbreaking nor particularly novel.**
>
> We have addressed this in the common questions section of the rebuttal.
>
> **Why was DeepSpeed Ulysses selected as the baseline? Does it exclusively support eager-mode and quadratic attention? A valid comparison would require using faster CP implementations, such as Megatron-LM's CP or Ring Flash Attention**
>
> DeepSpeed-Ulysses was selected as a baseline as it is the fastest CP parallel approach we have uncovered in the literature. See [1] Figures 9, 10 & 11 which compare DeepSpeed Ulysses and RingAttention across a variety of sequence-lengths, showcasing the Ulysses is faster than RingAttention on all tested configurations. However, during the rebuttal process, we have additionally benchmarked RingAttention in the same configuration as Figures 5 & 6 in our paper to demonstrate the trainability gains against it.
>
> Comparing RingAttention against AutoSP across various model-sizes on 8 GPUs (with an identical configuration to Figure 5 in our paper):
>
> | Model Size | AutoSP | RingAttention |
> |------------|--------|---------------|
> | 3B         | 75k    | 35k           |
> | 8B         | 45k    | 15k           |
> | 13B        | 15k    | 9k            |
>
> On average, AutoSP incurs a 2.26x average increase in training contexts across all model-sizes in these settings.
>
> Comparing RingAttention against AutoSP across different GPU counts (with an identical configuration to Figure 6 in our paper):
>
> | GPU Count | AutoSP | RingAttention |
> |-----------|--------|---------------|
> | 4         | 30k    | 12k           |
> | 8         | 90k    | 37k           |
>
> On average, AutoSP incurs a 2.45x average increase in training contexts across all model-sizes in these settings.
>
> Second, DeepSpeed Ulysses is implemented in eager-mode PyTorch, however, we have extended it to torch.compile for our evaluation. For all our benchmarked results (including both Eager Mode and torch.compile), the attention kernels are linear-memory flash-attention kernels.
>
> [1] https://dl.acm.org/doi/pdf/10.1145/3731569.3764798

---

> > ### Comment · Reviewer_AhGd · 2025-11-16
> >
> > Thank you for your careful answers. Additionally, I'd like to clarify that RingAttention is a very slow implementation of context parallelism, while Ring Flash Attention[1] is claimed to be significantly faster.
> >
> > [1] https://github.com/zhuzilin/ring-flash-attention

---

> > ### Comment · Reviewer_AhGd · 2025-11-16
> >
> > A key concern is that standard layer-wise gradient checkpointing already has a relatively low latency overhead (at 14-18%), and its simplicity suggests minimal room for further optimization. Therefore, it is reasonable to ask for empirical evidence demonstrating that your proposed nuanced checkpointing provides a non-trivial (or significant) advantage over this established baseline.

---

> ### Author Response · Authors · 2025-11-21
>
> **I'd like to clarify that RingAttention is a very slow implementation of context parallelism, while Ring Flash Attention is claimed to be significantly faster.**
>
> Thank you for your clarifications. To avoid confusion, RingAttention is known to be substantially slower due to its ring-style communication pattern. The more recent Ring Flash Attention  implementation (https://github.com/zhuzilin/ring-flash-attention) is faster, and all of our rebuttal results already use Ring-Flash-Attention, not the original RingAttention.
>
> To further address your concern, we conducted an additional analysis to compare AutoSP against Ring-Flash-Attention. We benchmark a Llama 3.1-8B model on 8 A100-80GB GPUs with sequence length 15k, SP size 2 and DP size 4. We measure the average latency of 10 training iterations after a warmup:
>
> | Technique                   | Runtime (s) | Memory Consumption (GB) |
> |-----------------------------|-------------|-------------------------|
> | Ring-Flash-Attention (15K)  | 5.29        | 72.6                    |
> | AutoSP - No AC pass (15k)   | 3.24        | 76                      |
> | AutoSP - with AC pass (15k) | 3.58        | 51                      |
>
> We observe that AutoSP is 1.44x faster with (and 1.62x faster without) its SP-aware AC-pass than Ring Flash Attention. With our SP-aware AC-pass enabled, AutoSP also uses 1.42x less memory than Ring-Flash-Attention.
>
> *Why does AutoSP outperform Ring Flash Attention?* The difference comes from how the communication pattern matches the hardware topology. RingAttention relies on a p-stage neighbor passing pattern, where each device repeatedly sends  [N/p] sized messages to its neighbour. This serialized ring pipeline is latency-bound and underutilizes the NVSwitch/NVLink crossbar. In contrast, AutoSP’s Ulysses-style SP uses two all-to-all collectives, which modern GPU systems have heavily optimized. In particular, AutoSP partitions the data between the two layouts: [b, N/p, h] & [b, N, h/p] (b, N, h, p are the batch, sequence length, hidden dimension and number of processes respectively). This leads to sending and receiving message sizes: [N/p] per link/device. Since NVSwitch is designed exactly for many-to-many patterns, it allows all-to-all communication to run fully in parallel across the NVLinks. As a result, AutoSP benefits from parallel bandwidth, whereas RingAttention suffers from p sequential hops and additional forward overhead.
>
> **A key concern is that standard layer-wise gradient checkpointing already has a relatively low latency overhead (at 14-18%), and its simplicity suggests minimal room for further optimization. Therefore, it is reasonable to ask for empirical evidence demonstrating that your proposed nuanced checkpointing provides a non-trivial (or significant) advantage over this established baseline.**
>
> Thank you for the suggestion. We would like to clarify that activation checkpointing is not the primary contribution of AutoSP. AutoSP’s core contribution is a compiler-driven automatic sequence parallelism (SP) transformation that lifts SP into the PyTorch-2 compilation pipeline through IR analysis and layout rewrites. This allows SP to be applied automatically to arbitrary user models without any manual code changes, which is something not supported by existing SP implementations.
>
> Activation checkpointing is only one component within AutoSP, which is used to ensure that the compiler-generated SP schedule remains memory-feasible for long sequences. Our goal is not to outperform standard layer-wise checkpointing on latency. Instead, the goal is to provide a fully automated SP-aware memory management strategy that integrates cleanly with the compiler. Traditional layer-wise AC via `torch.utils.checkpoint` indeed has a moderate latency overhead, but it requires user instrumentation of the model and runtime and therefore cannot be used in automated compiler passes.
>
> Within the space of automated AC solutions, the strongest baseline is PyTorch-2's built-in automated AC pass, which uses a min-max flow-based formulation to select checkpoints and is widely regarded as state-of-the-art for low-overhead automated AC. All of our ablations in Section 4.2 are run with this pass enabled. Despite this strong baseline, our SP-aware AC-pass increases the maximum training context by 1.6x while adding only 9% latency overhead on top of PyTorch-2’s automatic AC. Without our SP-aware AC, the compiler-generated SP schedule would run out of memory at very long sequence lengths.  This demonstrates that AutoSP’s memory component is effective, which enables the PyTorch-2 compiler to train long sequences that would otherwise be infeasible while keeping latency competitive with the best available automated AC baseline.

---

> > ### Author Response · Authors · 2025-11-23
> >
> > Dear Reviewer AhGd,
> >
> > Thank you once again for the time and effort you’ve invested in reviewing our manuscript. We would like to kindly remind you that we have diligently addressed each point raised in your review. We would be more than happy to address any additional concerns or comments you may have.
> >
> > Thank you!
> >
> > Best Regards,
> >
> > Authors of AutoSP (Submission 22044)

---

> > > ### Author Response · Authors · 2025-11-27
> > >
> > > Dear Reviewer AhGd,
> > >
> > > We hope you're doing well. We submitted our follow up responses on 11/21/2025, and we just wanted to gently check in since it has been 6 days and there are only a few days left in the rebuttal period.
> > >
> > > If there are any remaining concerns we can help clarify, we'd be very happy to provide additional explanations. Thank you again for the time and effort you've put into reviewing our paper.
> > >
> > > Best Regards,
> > >
> > > Authors of 22044

---

> > > ### Comment · Reviewer_AhGd · 2025-11-28
> > >
> > > Thanks for the response. After thinking it over, I think this work has some value regarding its engineering implementation. I am open to accepting this paper. I recommend that the authors submit a PR to PyTorch so the community can further verify this work.

---

### Official Review · Reviewer_SrRm · 2025-11-01

**Soundness:** 3
**Presentation:** 3
**Contribution:** 2
**Rating:** 6
**Confidence:** 3

**Summary:**

The paper introduces two compiler passes to achieve sequence parallel. It inserts all-to-all collectives and transform single GPU model into sequence parallelsized models. It also consider activation checkpointing that selectively rematerializes activations by finding best memory and perf trade offs. Similar to existing work like simpleFSDP and deepcompile, it capture the graph single GPU model and transform the IR with collectives and IR passes. It mains simple UX and single-GPU style authoring.

**Strengths:**

fair originality: this is indeed the 1st paper I saw to use compiler pass to achieve sequence parallel. DeepSpeed-Ulysses and RingAttention implement SP at the framework level that affects the single gpu authoring. This paper lifts SP into the compiler layer. Joint optimization with activation checkpointing is non-trivial and essential to make good memory perf trade offs.

enough quality: it touched the reasoning behind choosing which layer of IRs, and how collectives are inserted and optimized for better perf.   Evaluation is done on both NVIDIA and AMD GPUs, including eager and compiler baselines

descent clarify: The main technical sections (Sections 3.1–3.2) may challenge readers unfamiliar with PyTorch internals. Fortunately figures and intros effectively lowed the bar to keep up with the content.

highly significant: it's critical to showcase how to use IR passes to achieve SP. Long-context LLM training is a critical challenge. It's espectially benifitial to minimal developer friction. This could potentially influence future designs of pytorch compiler solution for parallelsims, like simpleFSDP

**Weaknesses:**

originiality mainly comes from using IR to implement SP. SP itself was previously implemented in open source library like DeepSpeed-Ulysses. There are also similar work that moves FSDP into IR passes. This paper is more like an extension of the idea to more parallelsims.

technical depth: it remains questionable how non-trivial it is to come up with IR passes to achieve SP, considering we have open source implementation in eager mode. The non-trivial evaluation should be done for people with enough understanding of pytorch 2 compiler stack. But I agree joint optimization with activation checkpoint is non-trivial

**Questions:**

Explain why it's non-trivial to come up with IR passes according to open source eager SP implementation like DeepSpeed-Ulysses

For maximum sequence length, analyze memory snapshot for each baseline and show more insights into memory usage: when did the peak happen, % of memory on model/opt state, activation, and intermidate tensors

---

> ### Author Response · Authors · 2025-11-16
>
> We thank you for your thoughtful reviews and are encouraged by your positive comments on how our technique is highly-significant, original, informative, and clearly communicated.
>
> **This paper is more like an extension of the idea to more parallelisms.**
>
> We have addressed this in the common questions section of the rebuttal.
>
> **Explain why it's non-trivial to come up with IR passes according to open source eager SP implementation like DeepSpeed-Ulysses.**
>
> We have addressed this in the common questions section of the rebuttal.
>
> **For maximum sequence length, analyze memory snapshot for each baseline: % of memory on model/opt state, activation, and intermediate tensors.**
>
> We benchmark DeepSpeed Ulysses (both eager mode and inductor compiled) and AutoSP on a Llama-3.1 8B model with an identical configuration as that in Figure 5 of our paper. We assess the amount of memory occupied by: activations, optimiser states and model parameters at the maximum trainable sequence length for each method. Below are our results:
>
> | Technique                            | Activation States (GB) | Optimiser States (GB) | Model Parameters (GB) |
> |--------------------------------------|------------------------|-----------------------|-----------------------|
> | AutoSP (45k)                         | 24.9                   | 8                     | 16                    |
> | DeepSpeed-Ulysses (Eager Mode - 10k) | 36.2                   | 8                     | 16                    |
> | DeepSpeed-Ulysses (Inductor - 15k)   | 37.6                   | 8                     | 16                    |
>
> We observe that AutoSP produces fewer activations compared to DeepSpeed-Ulysses despite being trained on larger sequence lengths. This is a result of the SP-aware AC-pass that releases the activation memory corresponding to linear-layers at negligible cost to training throughput.

---

> > ### Author Response · Authors · 2025-11-23
> >
> > Dear Reviewer SrRm,
> >
> > Thank you once again for the time and effort you’ve invested in reviewing our manuscript. We would like to kindly remind you that we have diligently addressed each point raised in your review. We would be more than happy to address any additional concerns or comments you may have.
> >
> > Thank you!
> >
> > Best Regards,
> >
> > Authors of AutoSP (Submission 22044)

---

> > > ### Author Response · Authors · 2025-11-27
> > >
> > > Dear Reviewer SrRm,
> > >
> > > We hope you're doing well. We submitted our rebuttal on 11/15/2025, and we just wanted to gently check in since it has been 12 days and there are only a few days left in the rebuttal period.
> > >
> > > If there are any remaining concerns we can help clarify, we'd be very happy to provide additional explanations. Thank you again for the time and effort you've put into reviewing our paper.
> > >
> > > Best Regards,
> > >
> > > Authors of 22044

---

### Official Review · Reviewer_K3Y6 · 2025-11-01

**Soundness:** 2
**Presentation:** 2
**Contribution:** 1
**Rating:** 2
**Confidence:** 4

**Summary:**

The paper introduces AutoSP, a compiler-based system designed to optimize the training of large language models (LLMs) for long-context scenarios. AutoSP automates sequence parallelism and activation checkpointing specifically tailored for long-contexts, aiming to enhance trainability without sacrificing runtime performance. The system is evaluated on both NVIDIA and AMD hardware, showing significant improvements in trainable sequence lengths compared to existing methods.

**Strengths:**

The paper presents a compiler-based solution for optimizing LLM training in long-context scenarios, which maybe easy to use.

**Weaknesses:**

1. Lack of novelty. Automatic parallelism has been thoroughly studied and there are lots of work about the automatic or dynamic sequence parallelism including selective activation checkpointing. This article integrates these elements into the compiler, which more like an engineering project.
 2.The paper's baseline comparisons may not fully represent the current state-of-the-art techniques, particularly in terms of hand-optimized implementations.
3.Lots of errors. Such as line 328 , “Grouped-Query-Attention (GQA) or Full-Attention” GQA and Full-attention are completely compatible.

**Questions:**

see weakness above.

---

> ### Author Response · Authors · 2025-11-16
>
> We thank you for your thoughtful reviews and are encouraged by your positive comment on how our compiler-based SP solution provides an easy-to-use methodology for long-context optimisations.
>
> **Automatic parallelism has been thoroughly studied with lots of work about automatic or dynamic sequence parallelism including selective activation checkpointing. In light of this, what is the novelty?**
>
> We have addressed this in the common questions section of the rebuttal.
>
> **The paper's baseline comparisons may not fully represent the current state-of-the-art techniques, particularly in terms of hand-optimized implementations.**
>
> We note that DeepSpeed-Ulysses is the SoTA long-context SP method. As demonstrated in figure 9 and 10 in [1], DeepSpeed-Ulysses is consistently faster than any other SP technique which is why we chose to evaluate against it. However, during the rebuttal we have additionally benchmarked the trainability of AutoSP against RingAttention. Our results are in the table below for convenience (we use the same configuration as Figures 5 & 6 in our paper).
>
> Comparing RingAttention against AutoSP across various model-sizes on 8 GPUs (identical configuration to Figure 5 in our paper):
>
> | Model Size | AutoSP | RingAttention |
> |------------|--------|---------------|
> | 3B         | 75k    | 35k           |
> | 8B         | 45k    | 15k           |
> | 13B        | 15k    | 9k            |
>
> On average, AutoSP incurs a 2.26x increase in training contexts across all model-sizes.
>
> Comparing RingAttention against AutoSP across different GPU counts (identical configuration to Figure 6 in our paper):
>
> | GPU Count | AutoSP | RingAttention |
> |-----------|--------|---------------|
> | 4         | 30k    | 12k           |
> | 8         | 90k    | 37k           |
>
> On average, AutoSP incurs a 2.45x increase in training contexts across all model-sizes.
>
> However, we note that RingAttention can be used when the SP group size exceeds the number of attention-heads, a scenario where Ulysses does not apply.
>
> [1] https://dl.acm.org/doi/pdf/10.1145/3731569.3764798
>
> **“Grouped-Query-Attention (GQA) or Full-Attention” GQA and Full-attention are completely compatible.**
>
> Whilst GQA and Full-attention incur the same FLOPs, they introduce different communication patterns with respect to context/sequence-parallelism. GQA, in reducing the number of k/v heads, changes the all-to-all prior/post the attention computation with respect to Full-attention (as the all-to-all changes the tensors from being partitioned across the sequence, to partitioned across the head-dimension). All these intricacies must be thought through when implementing a general-purpose SP compiler-pass.

---

> ### Author Response · Authors · 2025-11-23
>
> Dear Reviewer K3Y6,
>
> Thank you once again for the time and effort you’ve invested in reviewing our manuscript. We would like to kindly remind you that we have diligently addressed each point raised in your review. We would be more than happy to address any additional concerns or comments you may have.
>
> Thank you!
>
> Best Regards,
>
> Authors of AutoSP (Submission 22044)

---

> > ### Author Response · Authors · 2025-11-27
> >
> > Dear Reviewer K3Y6,
> >
> > We hope you're doing well. We submitted our rebuttal on 11/15/2025, and we just wanted to gently check in since it has been 12 days and there are only a few days left in the rebuttal period.
> >
> > If there are any remaining concerns we can help clarify, we'd be very happy to provide additional explanations. Thank you again for the time and effort you've put into reviewing our paper.
> >
> > Best Regards,
> >
> > Authors of 22044

---

### Author Response · Authors · 2025-11-16
**Responses to Common Questions**

We thank all reviewers for their thoughtful and constructive feedback. We are encouraged that reviewers highlighted the originality (SrRm, 4FNs), significant impact (SrRm, 4FNs), and ease of use (K3Y6, SrRm, 4FNs) of our method. We first address common questions raised by multiple reviewers, followed by responses to specific reviewer comments.

**Why AutoSP is novel**

We appreciate the reviewers’ questions. AutoSP is *the first system to lift sequence parallelism into the PyTorch-2's compilation stack via IR analysis and rewrites*. Implementing SP inside a compiler, rather than through eager-mode APIs, enables capabilities that existing SP libraries cannot provide:

1. Zero code changes for arbitrary PyTorch models. AutoSP automatically discovers attention/MLP structures in Torch-IR and inserts correct SP collectives. Eager SP (e.g., DeepSpeed-Ulysses, RingAttention) requires manual code refactoring, e.g., manual instantiation of SP groups, and sharding of input tensors to the training pipeline. This has to be done for every new framework and model that wishes to use SP.
2. Composable parallelism and optimizations (SP + FSDP + AC). Expressing SP a compiler pass allows it to compose cleanly with other compiler passes. For example, our SP-pass and AC-pass jointly optimize memory for long contexts.  On the other hand, implementing SP via an eager-mode implementation would require users to think through how to compose SP with other techniques.
3. Hardware-agnostic parallelization. Because AutoSP works at Pytorch’s middle-end IR, it automatically supports all PyTorch backends without per-backend engineering. In contrast, solutions like DeepSpeed-Ulysses only support a few attention backends, e.g., FlashAttention-v1/v2, Triton-FlashAttention.

**Why implementing SP in a compiler is non-trivial**

1. Implementing a compiler is substantially more challenging than implementing SP in eager mode:
Choosing the correct IR with correct semantics. PyTorch-2 has three distinct IRs (torch-IR, Aten-IR, Inductor-IR). Backend graphs are generated between the lowering of torch-IR to Aten-IR. Therefore, incorrect rewrites break autograd semantics. We designed AutoSP to rewrite at Torch-IR so that AOTAutograd generates a correct backward graph.
2. Joint forward/backward rewriting. Eager SP only inserts Python-level collectives in forward/backward functions. At the compiler level, we must make sure that IR rewrites in the forward graph automatically propagate to the generated backward graph without violating autograd invariants.
3. Robust pattern detection across arbitrary models. Program analysis must correctly identify which tensors to partition across the head-dimension (those in attention layers) and which tensors to partition across the sequence dimension (those in MLP layers). This is straightforward in eager mode, but non-trivial in fine-grained IR where the structure has already been transformed, e.g., flattened and fused.
4. Correct collective scheduling through other passes. Collectives must be inserted in IR positions that remain semantically correct after downstream passes (e.g., AOTAutograd graph generation, lowering,and our AC passes). Ensuring this invariance is non-trivial and requires significant efforts to maintain graph correctness.

These challenges are inherent to compiler design and are absent from eager-mode implementations. We will clarify this more explicitly in the revision.

---

### Comment · Area_Chair_kWkh · 2025-11-23
**Reviewer & Author Discussion**

Hi Reviewers,

Please kinly and actively participate in the review-author dicussion, raise your further concerns so that the authors can explain more, and make your final decisions.

---

### Meta-Review · Area_Chair_Uy7F · 2025-12-04

**Summary:**

The paper introduces AutoSP, a compiler-based system designed to optimize the training of large language models (LLMs) for long-context scenarios. AutoSP automates sequence parallelism and activation checkpointing specifically tailored for long-contexts, aiming to enhance trainability without sacrificing runtime performance. The system is evaluated on both NVIDIA and AMD hardware, showing significant improvements in trainable sequence lengths compared to existing methods.

**Reviewer Concerns:**

Two reviewers clearly reject it, while two reviewers agree to accept it.  The most concerns are on novelty and experiments. I think the rebuttals have partially addressed these concerns.

**Reviewer Scores:**

One reviewer has agreed to accept it after many round rebuttals. So I think this reviewer will increase his score.

---

### Decision · Program_Chairs · 2026-01-26

Accept (Poster)